# Specific NOX4 Inhibition Preserves Mitochondrial Function and Dampens Kidney Dysfunction Following Ischemia–Reperfusion-Induced Kidney Injury

**DOI:** 10.3390/antiox13040489

**Published:** 2024-04-19

**Authors:** Tomas A. Schiffer, Lucas Rannier Ribeiro Antonino Carvalho, Drielle Guimaraes, Ariela Boeder, Per Wikström, Mattias Carlström

**Affiliations:** 1Department of Physiology and Pharmacology, Karolinska Institutet, 17165 Solna, Sweden; tomas.schiffer@ki.se (T.A.S.); lucas.carvalho@ki.se (L.R.R.A.C.); drielle.braga@ki.se (D.G.); ariela.boeder@ki.se (A.B.); per.wikstrom@glucoxbiotech.com (P.W.); 2Department of Pharmacology, Federal University of Santa Catarina, Florianópolis 88040-900, Brazil; 3Glucox Biotech AB, 17997 Färentuna, Sweden

**Keywords:** ischemia–reperfusion, kidney, glomerular filtration rate, NOX4, mitochondria

## Abstract

**Highlights:**

**What are the main findings?**
NADPH oxidases (NOXs) are induced following ischemia-reperfusion (IR), which aggravates acute kidney injuryNOX4 inhibition improves kidney and mitochondria function in IR, and dampens renal injury

**What is the implication of the main finding?**
These findings can facilitate development of new preventive strategies of IR-induced AKI, which is a global medical problem.

**Abstract:**

**Background**: Acute kidney injury (AKI) is a sudden episode of kidney failure which is frequently observed at intensive care units and related to high morbidity/mortality. Although AKI can have many different causes, ischemia–reperfusion (IR) injury is the main cause of AKI. Mechanistically, NADPH oxidases (NOXs) are involved in the pathophysiology contributing to oxidative stress following IR. Previous reports have indicated that knockout of NOX4 may offer protection in cardiac and brain IR, but there is currently less knowledge about how this could be exploited therapeutically and whether this could have significant protection in IR-induced AKI. **Aim:** To investigate the hypothesis that a novel and specific NOX4 inhibitor (GLX7013114) may have therapeutic potential on kidney and mitochondrial function in a mouse model of IR-induced AKI. **Methods**: Kidneys of male C57BL/6J mice were clamped for 20 min, and the NOX4 inhibitor (GLX7013114) was administered via osmotic minipump during reperfusion. Following 3 days of reperfusion, kidney function (i.e., glomerular filtration rate, GFR) was calculated from FITC-inulin clearance and mitochondrial function was assessed by high-resolution respirometry. Renal histopathological evaluations (i.e., hematoxylin–eosin) and TUNEL staining were performed for apoptotic evaluation. **Results**: NOX4 inhibition during reperfusion significantly improved kidney function, as evidenced by a better-maintained GFR (*p* < 0.05) and lower levels of blood urea nitrogen (*p* < 0.05) compared to untreated IR animals. Moreover, IR caused significant tubular injuries that were attenuated by simultaneous NOX4 inhibition (*p* < 0.01). In addition, the level of renal apoptosis was significantly reduced in IR animals with NOX4 inhibition (*p* < 0.05). These favorable effects of the NOX4 inhibitor were accompanied by enhanced Nrf2 Ser40 phosphorylation and conserved mitochondrial function, as evidenced by the better-preserved activity of all mitochondrial complexes. **Conclusion**: Specific NOX4 inhibition, at the time of reperfusion, significantly preserves mitochondrial and kidney function. These novel findings may have clinical implications for future treatments aimed at preventing AKI and related adverse events, especially in high-risk hospitalized patients.

## 1. Introduction

Acute kidney injury (AKI) is a condition commonly seen in hospitalized patients. Mortality caused by AKI surpasses the collective burden imposed by diabetes, heart failure, or breast cancer [1]. Mortality has remained high in recent years [2]. With the ageing population, the incidence of AKI has increased in recent decades [3,4,5]. Causes of AKI include surgery, trauma, nephrotoxic drugs, kidney transplantation, heart disease, and sepsis [6]. Because there are currently no effective treatments, identification of new therapeutic strategies is of paramount importance.

Ischemia–reperfusion (IR) injury is the most important factor causing AKI [7]. During ischemia, the acidic state prevents mitochondrial permeability transition pore (mPTP) opening [8]. At reperfusion and reoxygenation, numerous oxygen free radicals are generated in complex I as a result of excessive succinate oxidation [9], leading to oxidative stress and oxidative damage to DNA, lipids, and proteins [10]. In addition, ROS generation during reperfusion promotes mPTP opening, which exacerbates IR damage [11]. Finally, danger-associated molecular patterns (DAMPs) are released that activate the innate immune response and trigger the release of cytokines, which promotes the recruitment of macrophages and neutrophils [12].

NADPH oxidase 4 (NOX4) is constitutively active and predominantly produces hydrogen peroxide [13]. It is mainly localized in mitochondria but also in the endoplasmic reticulum and nucleus [14,15,16,17]. Recently, it has been suggested that NOX4 serves as an energetic sensor following the demonstration that ATP can inhibit NOX4 activity [18]. It has also been reported to inhibit complex I activity in mitochondria [19]. Depending on the subcellular location, NOX4 regulates a variety of cellular functions such as TGF-β1-induced differentiation, transcriptional regulation, and cytoskeletal dynamics [20].

NOX4 is upregulated after IR injury [21,22], and several studies have shown that knocking out or downregulation of NOX4 leads to a reduction in infarct size after cardiac IR [21,23]. In addition, pharmacological inhibition of NOX4 has been shown to improve contractile function after cardiac IR [24]. The same protective effect was observed in ischemic stroke where NOX4 was identified as one of the main players in post-ischemic neurodegeneration [25,26]. Combined pharmacological NOX1/NOX4 inhibition has recently been shown to be protective also in septic AKI [27].

On the contrary, a report showed that both over expression and complete elimination of NOX4 in perfused hearts had a negative outcome where overexpression of dominant negative (non-functional) NOX4 contributed to increased reductive stress [28]. The contradictory findings may have been influenced by the fact that NOX4 activity was abolished already at the time of ischemia [29]. Interestingly, dual knockout of NOX2 and NOX4 enhanced cardiac injury after IR [21]. Low levels of ROS from NOX2/4 regulate Hif-1α, which confers protection [21], suggesting there is a therapeutic window in terms of ROS production. Moreover, findings suggest a cell type-specific function of NOX4 in ischemia, where an NOX4-dependent protective effect was observed in endothelial cells by promoting angiogenesis and attenuating inflammation [30].

The aim of this study was to pharmacologically inhibit NOX4, by using the novel and specific NOX4 inhibitor GLX7013114 (Glucox Biotech AB, Sweden), to investigate the hypothesis that during the reperfusion phase this treatment strategy could preserve mitochondrial and renal function in a mouse model of IR-induced acute kidney injury (AKI).

## 2. Methods

### 2.1. Ethics Statement

The experimental procedures received approval from the Institutional Animal Care and Use Committee in Stockholm (Approval Numbers: Dnr 17128-2021 and N139/15) and were conducted in compliance with the guidelines set forth by the US National Institutes of Health (NIH publication NO. 85-23, revised 1996) and the EU directive 2010/63/EU governing the execution of animal experiments. Detailed descriptions of the experimental methods utilized for the various in vitro, ex vivo, and in vivo studies are provided below.

### 2.2. NOX4 Inhibitor

The previously characterized specific NOX4 inhibitor GLX7013114 [31] was kindly provided by Glucox Biotech AB (Stockholm, Sweden).

### 2.3. Animals and Renal Ischemia–Reperfusion Model

Male C57BL/6J mice, 5 months old, were sourced from Janvier Laboratories (France) and kept in a controlled environment with a 12 h light/dark cycle. They were provided with standard rodent chow and tap water. It should be noted that some of the mice utilized in this study were also involved in a distinct yet parallel study. This approach aimed to minimize the number of animals used in accordance with the principles of the 3Rs. The model data, illustrating the impact of ischemia–reperfusion (IR) in comparison to sham-operated mice, have been partly reported in a recent study [32]. The animal research in this study was conducted and reported in accordance with the Arrive Guidelines Checklist.

Mice were anesthetized with isoflurane, and body temperature was maintained at 37 ± 0.5 °C throughout the procedure using a self-monitored heating pad combined with a heating lamp. Flank incisions were performed to enable access to both kidneys simultaneously. The kidneys were inspected and clamped for 20 min. Ischemia was confirmed by observation of renal color change. Mice were implanted with mini-osmotic pumps. For treatment during the first 24 h of the 3 d of reperfusion, mice received Alzet 2001D, 1 day (60 mg/mL), and for the 76 h treatment group, Alzet 2001, 7 days (420 mg/mL), was used containing the novel NOX4-specific inhibitor (GLX7013114, IC50: 0.3 µM). In addition, a naïve group of animals were subjected to NOX4i for 76 h. GLX7013114 was dissolved in 1:1 polyethylene glycol (PEG) and dimethyl sulphoxide (DMSO) (sham animals (Alzet 2001, 7 days) with solvent only). Animals were monitored and received buprenophrine (0.1 µg/g bw) 2 times daily for the first 2 consecutive days.

### 2.4. Glomerular Filtration Rate

The glomerular filtration rate (GFR) was estimated as previously described [32]. Briefly, FITC-inulin (TdB Labs, Uppsala, Sweden) was diluted in sterile PBS at a concentration of 10 mg/mL and filtered using a syringe filter (Sartorius AG, Minisart, Goettingen, Germany, 0.20 µm) prior to injection into the tail vein (100–200 µL). The syringe was weighed both before and after injection to determine the precise volume injected. Blood samples were taken from the tail tip at specified time points (1, 3, 5, 10, 15, 35, 55, 75 min) into pre-heparinized Eppendorf tubes. Following centrifugation, plasma was collected and shielded from light. Samples (3 µL) were diluted in PBS with 500 mM HEPES (67 µL). Fluorescence readings (excitation/emission 480/530 nm) were taken using a plate reader (Spectramax iD3, Molecular Devices). FITC-inulin clearance was determined using noncompartmental pharmacokinetic data analysis following the method detailed by Gabrielsson and Weiner [33]. Clearance was calculated as the administered intravenous dose of FITC-inulin divided by the total area under the plasma fluorescence time curve (AUC0-∞).

### 2.5. Histological Evaluation

Formalin-fixed kidney samples were dehydrated via incubation with increasing concentrations of ethanol, followed by diaffinization by xylol and embedding the sample in liquid paraffin. A microtome was used to cut the tissue blocks to a thickness of 5 µm. Hematoxylin–eosin (HE) and periodic acid–Schiff (PAS) stains were performed for evaluation under light microscopy. A minimum of 10 fields from the area of the tubular S3 segment (200× magnification) were scored semi-quantitatively by a blinded histopathologist regarding tubular injury. Tubular damage was characterized by the presence of tubular cast formation, tubular dilation, tubular atrophy, loss of the brush border or thickening of tubular epithelial cells, and thickening of the tubular basement membrane, as evaluated using the following scoring system: a Score of 0 indicated no tubular damage, 1 represented less than 10% of tubules being affected, 2 denoted 10–25% of tubules being affected, 3 showed 25–50% of tubules being affected, 4 implied 50–74% of tubules being affected, and 5 indicated more than 75% of tubules being affected [34]. The statistical analysis was carried out using the Kruskal–Wallis test.

### 2.6. Evaluation of Apoptosis in Kidney Tissue

A TUNEL-HRP-DAB assay kit (ab206386; Abcam) was utilized to evaluate cell apoptosis in the kidney. Following fixation in 4% formalin, kidney specimens were embedded in paraffin and sectioned into 5 μm slices. The assay was conducted in accordance with the manufacturer’s guidelines. To summarize, after deparaffinization and rehydration, the sections underwent permeabilization with proteinase K for 20 min. Subsequently, a 3% H_2_O_2_/methanol solution was applied for 5 min to inhibit endogenous peroxidase activity. The slides were then exposed to the TUNEL reaction mixture in a humidified chamber for 90 min, followed by a 10 min block with blocking buffer. The samples were treated with the streptavidin–horseradish peroxidase (HRP) conjugate, diluted at a ratio of 1:25, for 30 min. Diaminobenzidine (DAB) was introduced, while methyl green served as a counterstain. Imaging was conducted using a light microscope (ZEISS Axioscope, Oberkochen, Germany) equipped with a 20× objective. Quantitative analysis was executed employing the particle analysis function in ImageJ software to determine the TUNEL-positive area percentage.

### 2.7. Mitochondrial Isolation

On the third day of reperfusion, kidney tissues were harvested, and mitochondria were isolated through a process of differential centrifugation, as described elsewhere [35]. In summary, the tissue underwent homogenization using a glass homogenizer (Potter Elvehjem) in an isolation medium composed of the following in mM: 250 sucrose, 10 Hepes, 1 EGTA, BSA 1 g/L, with pH 7.4 adjusted with KOH. Subsequently, the homogenate was subjected to centrifugation at 700× *g* for 10 min. The resulting supernatant was collected and centrifuged once more at 10,000× *g* for 10 min. The resultant pellet was meticulously washed, and the buffy coat was removed. It was then resuspended in isolation buffer. The suspension underwent centrifugation at 7000× *g* for 5 min, followed by an additional washing step. The pellet was reconstituted in mitochondrial preservation medium (in mM: 3 MgCl_2_-6H_2_O, 20 taurine, 60 K-lactobionate, 2 malate, 110 sucrose, 20 histidine, 2 Mg-ATP, 2 glutamate, 5 EGTA, 10KH_2_PO_4_, 20 HEPES, 3 glutathione, 1 g L^−1^ BSA, 20 µM vitamin E succinate and 1 µM leupeptin) and allowed to stabilize for a minimum of 30 min before respiratory analysis. The pellet was reconstituted at a ratio of 1 μL preservation medium per milligram of initial sample weight.

### 2.8. Mitochondrial Function

Mitochondrial function was assessed using high-resolution respirometry (O2-K, Oroboros, Austria). The Respiratory Control Ratio (RCR) was calculated as the maximal respiratory capacity mediated by complex I (using pyruvate (5 mM), malate (2 mM), and ADP (2.5 mM)) divided by the leak state without adenylates (pyruvate, malate). The respiration medium contained in mM: 110 Sucrose, 10 KH_2_PO_4_, 20 HEPES, 3 MgCl_2_ × 6 H_2_O, 60 K-lactobionate, 20 Taurine, 0.5 EGTA, with pH 7.1. Maximal complex activities were measured in the presence of substrates as follows: Complex I (CI) with pyruvate (5 mM), malate (2 mM), and ADP (2.5 mM); Complex II (CII) with succinate (10 mM), ADP, and rotenone (0.5 µM); CI + CII, which largely resembled CIII activity, which is the enzymatic bottleneck in this condition [36], with pyruvate, malate, succinate, and ADP; Complex IV (CIV) with ascorbate (2 mM), TMPD (0.5 mM), antimycin (2.5 µM), ADP, and cytochrome c (10 µM). Corrections were made for autooxidation of TMPD/ascorbate under the given oxygen tension. Proton leak dependent respiration was measured in presence of: pyruvate, malate and oligomycin (10 nM). Mitochondrial H_2_O_2_ production was measured spectrofluorometrically (O2k-Fluo, Oroboros) during leak respiration by using the amplex red system (5 μM, Amplex ultrared^TM^) in the presence of horseradish peroxidase (1 U/mL) (Sigma-Aldrich P 8250). H_2_O_2_ signal was calibrated by the addition of a standard solution of hydrogen peroxide (180 nM) prior to each experiment. Respiration was normalized to tissue wet weight and mitochondrial protein.

### 2.9. Kidney Tissue Citrate Synthase Activity

Citrate synthase (CS) activity was assessed using a commercial colorimetric kit (Sigma-Aldrich, Burlington, MA, USA, MAK193). Kidney tissue was cryo-grinded in a mortar placed in liquid nitrogen, diluted in the provided assay buffer, and thereafter, the manufacturer’s protocol was adhered to. CS activity was normalized to protein content and reported as U/g protein.

### 2.10. Immunoblotting

Kidney tissue samples were cryo-grinded in a mortar placed in liquid nitrogen and then solubilized in RIPA buffer containing protease and phosphatase inhibitor cocktail (Sigma-Aldrich, Burlington, MA, USA, P6726, P8340). For HK2 cells, wells were washed twice with ice-cold PBS, harvested by a cell scraper, and resuspended in PBS. After centrifugation, cells were resuspended in RIPA buffer containing protease and phosphatase inhibitor cocktail. Samples were centrifuged at 15,000× *g* for 15 min and diluted in commercial Laemmli sample buffer (Bio-Rad, 1:4) supplemented with 10% 2-mercaptoethanol, followed by protein denaturation at 95 °C for 5 min. Proteins were separated by electrophoresis on precast gels (Bio-Rad, Hercules, CA, USA, Criterion™ TGX™, 4–20%). The Bio-Rad, Criterion™ Blotter was used to transfer proteins to polyvinylidene difluoride membranes at 350 mA for 55 min (20% methanol). Incubation with primary antibodies was performed overnight at 4 °C followed by incubation with secondary antibodies for 1.5 h at room temperature. The primary antibodies used were mouse anti-OXPHOS antibody cocktail (Thermofisher Scientific, Waltham, MA, USA, 45-8099), rabbit anti-NOX4 antibody (Novus Biologicals, Shanghai, China, NB110-58849SS), rabbit anti phospo-Nrf2 (Ser40) (Thermofisher, Thermofisher Scientific, Waltham, MA, USA, PA5-67520), mouse anti-*alpha tubulin* (*Novus* Biologicals, *NB100-690*). The secondary HRP-linked antibodies were anti-rabbit (7074S) and anti-mouse (7076S) (Cell signaling Technology, Danvers, MA, USA). For molecular weight estimation, protein standards were used (Precision Plus Protein™ Standards, Dual Color, Bio-Rad). Bands were visualized in the ChemiDoc MP Imaging System (Bio-Rad) by using SuperSignal West Femto Maximum Sensitivity Substrate (Thermofisher Scientific, Waltham, MA, USA). Proteins were normalized to either alpha-tubulin or total protein (Pierce™ Reversible Stain Kit, Thermo Scientific, Waltham, MA, USA) by dividing the density of the target band with the density of the loading control or the sum of band densities in case of normalization to total protein. Band density was analyzed by using the software Image Lab 6.0.1 (Bio-Rad). When analyzing the same protein ((P)-Nrf2, total protein) using two blots, a control sample from the same animal * was included in both gels. Given that the transfer efficiency of the control sample was assumed to be consistent with the rest of the samples on membrane 2, correction factors were implemented across the gels for both protein-specific and total protein measurements. These corrections were subsequently utilized for densitometric adjustments in the remaining samples.

### 2.11. Cell Culture

Human kidney 2 (HK-2) cells, an immortalized proximal tubule epithelial cell line, were seeded on 6-well plates and cultured at 37 °C (5% CO_2_) in DMEM/F12 (Gibco) [+] L-glutamine [+] 15 mM HEPES, 10% FBS and 100 U/mL penicillin, and 100 μg/mL streptomycin.

### 2.12. Hypoxia Reoxygenation Protocol in HK-2 Cells

The HK-2 cell hypoxia reoxygenation model (H/R) used in this study was slightly intensified (0.2% O_2_) compared to previous studies using 1% O_2_ for 12 h [37] as no significant effects on viability and mitochondrial ROS production were detected using the commonly used model. Cells were grown at 24-well plates and washed twice in tyrode buffer (in mM; 130 mM NaCl, 5 mM KCl, 10 mM Hepes, 1 mM MgCl_2_, 1.8 mM CaCl_2_) and maintained in tyrode buffer while placed in a hypoxic chamber (Whitley H35 hypoxystation) at 0.2% O_2_, 5% CO_2_, and 37 °C for a period of 3 h. The medium was exchanged to complete medium right before reoxygenation.

### 2.13. Mitochondrial ROS Production in HK-2 Cells

At 24 h of reoxygenation, MitoSOX™ was added to the cell medium according to the manufacturer’s instructions (5 µM MitoSOX). Ten minutes later, the cells were washed in PBS and observed in a widefield fluorescence microscope (Zeiss Cell Observer, Oberkochen, Germany). Fluorescence intensity was analyzed in Image J software (ImageJ, U. S. National Institutes of Health, Bethesda, MD, USA). To correct for variations in cell density, the image was thresholded to generate a mask defining the cellular area where the mean fluorescence intensity was measured. Background fluorescence was subtracted from the mean intensity. Cell viability was evaluated by trypan blue. The results were validated across three separate days of experimentation.

### 2.14. NOX Activity in HK-2 Cells

After 12 h of reoxygenation, the cells were washed twice with ice cold PBS and harvested with a cell scraper. Cells were disrupted by sonication on ice (2 × 10 s, MSE Soniprep 150). NOX activity was determined using a luminometer (Berthold Autolomat Plus LB 953). Luminol (10 µM) and horse radish peroxidase (HRP) (10 µg/mL) were added to the cell suspension. Catalase (500 U/mL) was added to distinguish between H_2_O_2_-dependent and -independent oxidation of luminol. For NOX2 activity, lucigenin (50 µM) was added to intact cells. The reaction was initiated by injection of NADPH (100 µM). NOX4 inhibitor was added to the cell suspension (2 µM) to evaluate the contribution of H_2_O_2_ production of NOX4. The chemiluminescence was measured for 3 min. Samples were analyzed using the software Berthold version 1.0. The signal was corrected for the blank sample. The signal was normalized to cellular protein content determined by a commercial BCA kit (Micro BCA™ Protein Assay Kit, Thermofisher Scientific). The findings were confirmed through experimentation on three distinct days.

### 2.15. Blood Urea Nitrogen

Blood urea nitrogen (BUN) was evaluated by using a commercial colorimetric kit by following the manufacturer’s instructions (Urea Nitrogen Colorimetric Detection Kit, EIABUN, Thermofisher Scientific).

### 2.16. Statistical Analysis

Statistical analysis involved the utilization of One-Way ANOVA followed by Tukey’s multiple comparisons test to identify significant variances between treated and untreated groups (conducted on GraphPad Prism 9.0). For non-parametric histopathological data, the Kruskal–Wallis test was employed. The results are expressed as means with standard deviations (± SDs). *p* < 0.05 was considered significant. Symbols were used to denote significance levels as follows: * for *p* ≤ 0.05, ** for *p* ≤ 0.01, *** for *p* ≤ 0.001, and **** for *p* ≤ 0.0001. Non-significant variations were indicated as n.s.

## 3. Results

### 3.1. Kidney Function

Glomerular filtration rate (GFR) was measured for the evaluation of kidney function. The untreated group exposed to IR showed a significant reduction in GFR compared to the sham group (Figure 1A). While treatment with the NOX4 inhibitor during the first 24 h (of 3 d reperfusion) did not contribute to a significantly improved GFR, the group treated for 3 consecutive days had a significantly higher GFR compared to the untreated group (Figure 1A). The GFR data were supported by significantly increased blood urea nitrogen in the untreated group, which was significantly suppressed in the 3-day treated group (Figure 1B).

### 3.2. Histopathological Evaluation

Histopathological evaluation was performed on paraformaldehyde-fixed kidney slices. Tubular necrosis, dilatation, cast formation, and loss of brush border in the S3 segment was significantly reduced in the 76 h NOX4 inhibitor-treated animals compared to the untreated animals (Figure 2A,B).

### 3.3. Evaluation of Apoptotic Tissue

Apoptosis is a crucial part of the pathogenesis of acute kidney injury [38]. Surprisingly, results from the TUNEL staining suggested that NOX4 inhibitor treatment during the first 24 h of the 3 days of reperfusion was sufficient to protect against AKI-induced apoptosis, similar to continuous treatment over 76 h (Figure 3A,B).

### 3.4. Mitochondrial Function

Mitochondrial function was evaluated by high-resolution respirometry. Treatment with the NOX4 inhibitor preserved the mitochondrial degree of coupling reflected by the respiratory control ratio (RCR) (Figure 4A). Upon washing out the NOX4i during the mitochondrial isolation protocol, the mitochondrial H_2_O_2_ production during leak respiration was not significantly affected in any of the groups (Figure 4B). Moreover, intrinsic mitochondrial complex activities were preserved in both the group treated with NOX4i during the first 24 h and the whole reperfusion period of 76 h (Figure 5A–D). NOX4i inhibition during the whole reperfusion period contributed to significantly higher CIV activity, even surpassing the sham group (Figure 5D). 

### 3.5. Tissue Citrate Synthase Activity

CS activity is commonly considered to reflect mitochondrial content. Ischemia and 3 days of reperfusion did not affect CS activity, nor did NOX4 inhibition for 3 days (Figure 6).

### 3.6. Protein Levels of Components of the Mitochondrial Respiratory Chain

Immunoblotting targeting mitochondrial respiratory complexes (Figure 7A,B) indicated that ischemia and 3 days of reperfusion did not significantly affect the protein levels of CI, CII, CII, and CIV (Figure 7C–F).

### 3.7. Nrf2 Serine Residue 40 Phosphorylation

Apart from its well-established role in the response to oxidative stress, Nrf2 is increasingly being recognized in the context of preserving mitochondrial health [39,40]. Recently, a relation between NOX4 activity, Nrf2, and mitochondrial function has emerged [20,41]. Serine residue 40 phosphorylation is a requirement for nuclear translocation of Nrf2 [42]. To evaluate activation of Nrf2, immunoblotting of Ser40 phosphorylated Nrf2 was performed. On the third day of reperfusion, the levels of phosphorylated Nrf2 normalized to Nrf2 were significantly reduced in the IR group compared to the sham-operated mice. Treatment with NOX4i for both 24 h and 76 h of the 3 days of reperfusion prevented a reduction in Nrf2 levels following IR (Figure 8A,B). Similarly, Phospho-Nrf2 normalized to total protein was significantly improved in the 76 h NOX4i group compared to the untreated group (Figure 8C). The same trend was clearly observed in the 24 h NOX4i-treated group but did not reach statistical significance (Figure 8C). AKI contributed to reduced overall protein levels of Nrf2, which was significantly increased in the 76 h NOX4i-treated group (Figure 8D).

### 3.8. Mitochondrial ROS Production in HK-2 Cells Exposed to Hypoxia Reoxygenation

The effect of NOX4 inhibition on mitochondrial ROS production and viability in HK-2 cells after hypoxia reoxygenation (H/R) was evaluated in a cell model of IR. Mitochondrial ROS production increased significantly in HK-2 cells exposed to 3 h hypoxia (0.2% O_2_) and 24 h reperfusion (Figure 9A,B). ROS production was significantly reduced in cells treated with NOX4i (0.5 µM) compared to untreated cells (Figure 9A,B). Treating cells with the higher concentration of the inhibitor (2 µM) contributed to a paradoxical increased ROS production compared to control cells and was not significantly different to the untreated cells (Figure 9A,B). Similarly, viability was significantly improved in the cells treated with the 0.5 µM NOX4 inhibitor compared to untreated cells (Figure 9C). The cells treated with the 2 µM NOX4 inhibitor were not significantly different from either the untreated hypoxia reoxygenation exposed cells or normoxic cells (Figure 9C).

### 3.9. NOX4 Expression in HK-2 Cells Exposed to Hypoxia Reoxygenation

Immunoblotting targeting NOX4 (Figure 10A) was performed in cells exposed to hypoxia reoxygenation (H/R). Immunoblotting revealed significantly higher NOX4 protein levels compared to control cells (Figure 10B).

### 3.10. The Relative Contribution of H_2_O_2_ Production from NOX4 in HK-2 Cells Exposed to Hypoxia Reoxygenation

NOX activity was evaluated in cells exposed to hypoxia reoxygenation (H/R) combined with NOX4 specific inhibition. H/R contributed to significantly increasing both superoxide (Figure 11A) and hydrogen peroxide (Figure 11B) production. While NOX4-specific inhibition did not reduce superoxide production in whole cells (Figure 11A), the H/R- induced H_2_O_2_ production was prevented by NOX4 inhibition in lysed cells, suggesting that NOX4 was responsible for the main part of the increased H_2_O_2_ production (Figure 11B). The signal was completely prevented in presence of catalase, suggesting the high specificity of luminol for hydrogen peroxide in this assay.

## 4. Discussion

Acute kidney injury (AKI), defined as a rapid decline in kidney function (GFR) resulting in the retention of nitrogenous wastes and various complications, is a global health problem that affects millions of people each year [43]. There have been tremendous research efforts, both preclinical and clinical, trying to identify the causative factors underlying the development and progression of the disease but there are still many gaps to fill [44]. Regarding the underlying mechanisms, numerous factors/organs have been proposed to be involved, including abnormal redox balance and deficient nitric oxide signaling, as well as pro-inflammatory processes. Specifically, NADPH oxidases, and their interaction with the mitochondria [45], should be mentioned, as they are generally considered as major producers of reactive oxygen species (ROS) and have been involved in both renal and cardiovascular disorders, and therapeutic strategies damping their activity may have potential benefits [46]. Here, we investigated the therapeutic potential of a novel and specific NOX4 inhibitor during the reperfusion phase(s) following ischemia of the kidneys.

Remarkably, in our IR-induced AKI model, NOX4 inhibition during the reperfusion phase contributed to significant preservation of GFR (Figure 1), a better histopathological outcome in the S3 segment (Figure 2), preserved mitochondrial function (Figure 4), and reduced levels of apoptosis (Figure 3). In support of the improved GFR, BUN levels, in mice with IR plus the NOX4 inhibitor, were significantly suppressed after 3 days of consecutive treatment (Figure 1).

Tubular injury is a decisive step and occurs early in AKI [47] which contributes to several deleterious effects including impaired NaCl reabsorption [48]. As a consequence, it is generally believed that the increased load of NaCl at the macula densa in turn lowers GFR via activation of the tubuloglomerular feedback to inhibit urinary loss of NaCl [48]. Restoration of tubular integrity is therefore a prerequisite for GFR recovery. Indeed, NOX4 inhibition at reperfusion reduced tubular injury (Figure 2), which was linked to a better preserved GFR (Figure 1A) and attenuation of BUN accumulation (Figure 1B).

The in vivo results are in line with previous reports showing that knockout or downregulation of NOX4 offers cardio protection in a myocardial IR model [21,23], where over-expression of NOX4 aggravated ischemic injury ex vivo in isolated perfused hearts [28]. While we did not show the site and subcellular localization of NOX4 induction, previous studies have proven the induction of NOX4 at the location of renal injury [49,50].

Importantly, the novel NOX4-specific inhibitor preserves NOX2-dependent ROS production, where low levels of oxidative stress have been shown to mediate essential adaptive mechanisms through activation of Hif1α and suppression of PPAR α [21].

IR-induced apoptosis has been linked to loss of organ function in AKI [38]. Intriguingly, an anti-apoptotic effect was observed when treating the animals with the NOX4 inhibitor both during the first 24 h and the whole reperfusion period (Figure 3). Similarly, despite the lack of a protective effect on kidney function after NOX4 inhibition during the first 24 h, it was sufficient to largely preserve mitochondrial function, which also correlated with the observed reduced apoptosis. Mitochondria are crucial players in apoptotic signaling [51]; hence, the preserved mitochondrial function may have contributed to the anti-apoptotic effect. Even more surprising, CIV activity surpassed even the sham group after 3 days of treatment and trends in the same direction in naïve animals treated with NOX4i as well.

Silencing NOX4 has been found to unexpectedly boost mitochondrial biogenesis, increase complex I and II activity in lung fibroblasts, and activate Nrf2 via the PGC-1 alpha independent pathway [20]. Additionally, NOX4 inhibition in human beta-cells enhanced mitochondrial function [41], as evidenced by increased maximal oxygen consumption rates during administration of the uncoupling agent carbonyl cyanide-p-trifluoromethoxyphenylhydrazone (FCCP) in whole islets. Although no investigation has measured specific complex activities, Nrf2 is increasingly acknowledged for its role in maintaining mitochondrial health [39,40]. Preservation of mitochondrial function in NOX4i-treated animals may result from Nrf2-dependent mitochondrial biogenesis in the kidney. Indeed, immunoblotting revealed induced Nrf2 Serine-40 phosphorylation in the NOX4i-treated animals compared to the untreated IR group (Figure 8). Serine-40 phosphorylation is required for Nrf2 stabilization and translocation to the nucleus [42]. Although actual Nrf2 translocation to the nucleus at the site of injury was not confirmed in the present study, the results may at least in part explain the observed preservation of the mitochondrial function in the NOX4i-treated animals.

Nrf2 activation by silencing NOX4 is somewhat counterintuitive as Nrf2 usually escapes degradation upon Keap1 oxidation [52]. In contrast, several reports implicate NOX4 as a positive regulator of Nrf2, e.g., acute exercise activates Nrf2 in an NOX4-dependent manner [53,54]. However, Nrf2 translocation is regulated by Keap1-independent pathways [55,56,57], and it also protects mitochondrial function non-transcriptionally by associating with the outer mitochondrial membrane [58]. Another potential mechanism involves reduced NOX4 activity leading to increased nitric oxide bioavailability [23], known to enhance Nrf2 translocation through MAPK [59,60].

On the other hand, CS activity and the levels of mitochondrial proteins in the respiratory chain were unaffected by the ischemic insult after 3 days of reperfusion in our study. CS activity has recently been shown to be a poor marker of mitochondrial density in the kidney [61]. In a pathological situation, CS activity has frequently been shown to differ in relation to mitochondrial respiratory activity [62]. Our findings contradict previous in vitro work by Cho et al. that used MitoTracker as a marker of mitochondrial content [63]. The accumulation of MitoTracker within the mitochondrial matrix relies on the functionality of mitochondria with an intact membrane potential. In cases of dysfunctional mitochondria, the signal may be altered.

Still, the intact levels of the respiratory components in the untreated group indicate that the respiratory function rather than the mitochondrial content was reduced. A loss of mitochondrial complex activities in IR is a known feature and has been reported in several previous studies [64,65,66,67,68,69]. Recent reports show that a loss of flavin mononucleotide in complex I alters the activity without affecting the protein content in brain IR [70,71]. In our study, the underlying mechanism(s) that contributed to improved complex IV activity compared to the sham group in the 3-day NOX4-inhibited group are elusive and remain to be investigated.

Equal in vivo administration of GLX7013114 via minipump has, by the manufacturer, been shown to provide a plasma concentration of approximately 0.9 µM during the first hour and around 0.5 µM after 6 h (unpublished data), which fairly corresponds to the protective effect observed in the HK2 cells. The NOX4 IC50 of GLX7013114 is 0.3 µM. The absence of a protective effect in the cells at the higher concentration (2 µM) is indicative of a therapeutic window of ROS production as previously demonstrated, where combined NOX2/4 knockout had a detrimental effect on IR injury [21]. Profound inhibition of NOX4 may have led to the induction of reductive stress, which in turn can contribute to a paradoxical increased mitochondrial ROS generation through direct transfer of electrons to oxygen in the mitochondrial ETC [8,28,41].

In line with previous studies [21,22], the protein levels of NOX4 increased in HK2-cells exposed to H/R, which was supported by the increased measured NOX activity (Figure 11). Superoxide production increased in whole cells, which mainly reflects membrane-bound NOX2-derived superoxide production, which indicates that NOX2 was induced as expected [22]. Indeed, NOX4 inhibition in this experimental in vitro setting did not affect the signal. When using luminol as a H_2_O_2_ probe in lysed cells, a profound reduction in H_2_O_2_ production was observed upon NOX4 inhibition, suggesting that the increase was NOX4-mediated, confirming that the NOX4-specific inhibitor worked as expected.

In conclusion, chronic pharmacological inhibition of NOX4 contributes to preserved mitochondria and kidney function in a mouse model of IR-induced kidney injury. Thus, pharmacological NOX4 inhibition has the potential to reduce the risk of developing AKI, in the setting of IR, which is a global medical problem, especially in high-risk patients that undergo major surgeries and transplantations.

## Figures and Tables

**Figure 1 antioxidants-13-00489-f001:**
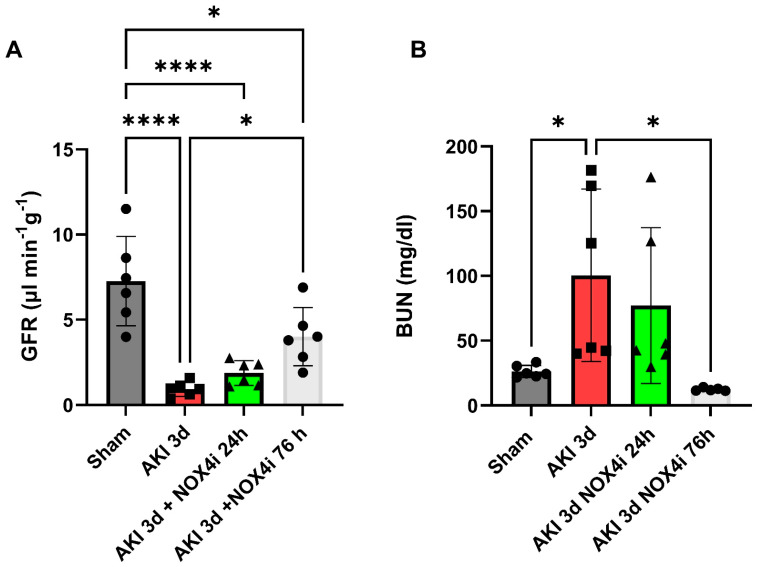
(**A**) Glomerular filtration rate (GFR) (N = 6/group) and (**B**) blood urea nitrogen (N = 6/group) was evaluated in mice treated with NOX4 inhibitor (NOX4i) during the first 24 h and during the whole reperfusion period of 3 days of reperfusion following 20 min bilateral ischemia (AKI). Venous injection of FITC-inulin was performed followed by sequential sampling of plasma. Plasma clearance of FITC-inulin was evaluated spectrofluorometrically. BUNs were measured by using a commercial colorimetric detection kit. One-Way ANOVA was used for statistical analysis. * for *p* ≤ 0.05, and **** for *p* ≤ 0.0001.

**Figure 2 antioxidants-13-00489-f002:**
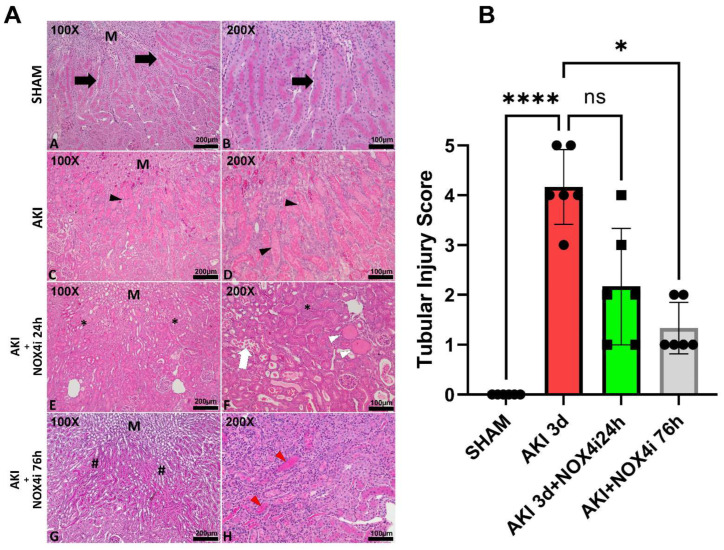
Histopathological evaluation of kidneys in the cortico–medullary junction (S3 segment area) in animals exposed to bilateral ischemia (20 min), treated with NOX4 inhibitor (NOX4i) during the first 24 h or 76 h of the 3-day reperfusion phase (N = 6/group). (**A**) Representative images of paraformaldehyde-fixed kidney slices stained with hematoxylin–eosin (HE) and periodic acid-Schiff (PAS) analyzed under light microscopy. ((**A**), *A–B*) Control group (SHAM) with normal tubular appearance, preserved cell morphology, and presence of an intact brush border positively highlighted by PAS (black arrow), in the region close to the renal medulla (M). ((**A**), *C–D*) Samples from the experimental group exposed to ischemia–reperfusion-induced kidney injury (AKI) showing extensive dilation of the S3 segment area tubules, cast formation, and sloughing of tubular epithelial cells or loss of the brush border (black arrowhead). ((**A**), *E–F*) Samples from the group exposed to AKI treated with specific NOX4i during the first 24 h of reperfusion showing a non-significant reduction in tubular damage in the S3 region close to the renal medulla (M), the majority of tubules with a normal diameter and cellular integrity (*), and other tubules presenting cellular debris in the lumen (white arrow) and the formation of hyaline casts (white arrowhead). ((**A**), *G–H*) Samples from the group with specific NOX4i for 76 h, with a significant reduction in tubular alterations, observed by the reduction in tubular dilation and greater integrity of cells and the brush border (#), with some tubules showing atrophy and the formation of hyaline casts (red arrowhead). (**B**) Statistical analysis of tubular injury score (Kruskal–Wallis test). Score 0: no tubular injury; Score 1: <10% of tubules injured; Score 2: 10–25% of tubules injured; Score 3: 25–50% of tubules injured; Score 4: 50–74% of tubules injured; Score 5: >75% of tubules injured. * for *p* ≤ 0.05, and **** for *p* ≤ 0.0001. Non-significant variations are indicated as ns.

**Figure 3 antioxidants-13-00489-f003:**
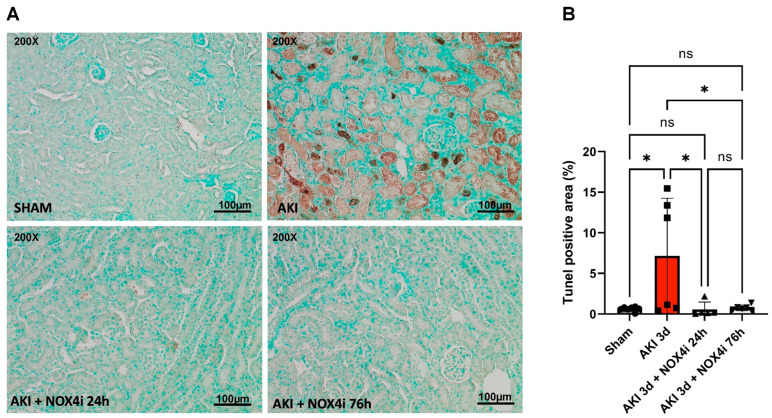
**A** TUNEL assay was performed in kidney slices to evaluate apoptosis in the kidney S3 segment following acute kidney injury (AKI) (N = 6/group) 20 min bilateral ischemia followed by 3 days of reperfusion) combined with NOX4 inhibition (NOX4i) during either the first 24 h or the whole 76 h of reperfusion. (**A**) Representative TUNEL-stained kidney slices, where brown areas represent apoptotic tissue. (**B**) The percentage of TUNEL-positive area on the 3rd day of reperfusion was estimated by using the particle analysis command in the ImageJ software. One-Way ANOVA was used for statistical analysis. * for *p* ≤ 0.05. Non-significant variations are indicated as ns.

**Figure 4 antioxidants-13-00489-f004:**
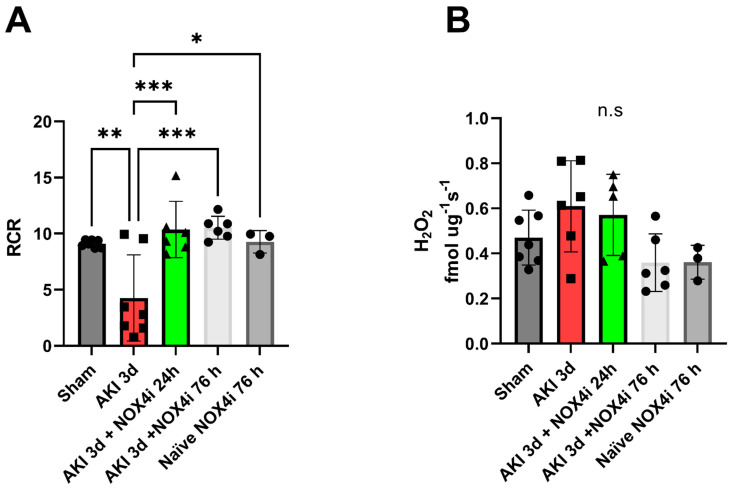
The degree of mitochondrial coupling and hydrogen peroxide production were evaluated on the 3rd day of reperfusion in isolated mitochondria from kidneys in mice following induced acute kidney injury (AKI) (20 min bilateral ischemia) combined with NOX4 inhibition (NOX4i). (**A**) The mitochondrial respiratory control ratio (RCR) defined as maximal CI-dependent state 3 respiration related to state 2 respiration in absence of adenylates was evaluated by high-resolution respirometry (Sham N = 7, AKI 3d and AKI 3d + NOX4i 76 h N = 6, NOX4i 24 h N = 5 and naïve + NOX4i 76 h N = 3) (**B**). Mitochondrial hydrogen peroxide (H_2_O_2_) production was measured spectrofluorometrically during leak respiration by using the amplex red system (N = 6/group, except for NOX4i 24 h N = 5). Note that NOX4i was washed away during the mitochondrial isolation protocol during these experiments. * for *p* ≤ 0.05, ** for *p* ≤ 0.01, and *** for *p* ≤ 0.001. Non-significant variations are indicated as ns.

**Figure 5 antioxidants-13-00489-f005:**
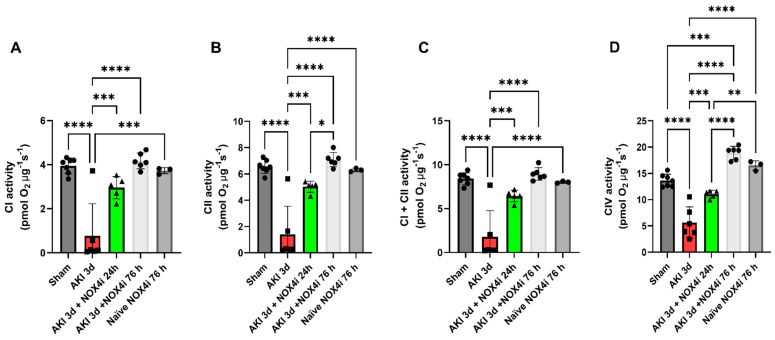
Kidney mitochondrial respiratory complex activities were evaluated by high-resolution respirometry on the 3rd day of reperfusion in isolated mitochondria from mice following induced acute kidney injury (AKI) (20 min bilateral ischemia) combined with NOX4 inhibition (NOX4i) during the first 24 h or during the whole reperfusion period (Sham N = 7, AKI 3d and AKI 3d + NOX4i 76h N = 6, NOX4i 24 h N = 5 and naïve + NOX4i 76 h N = 3). A control group of naïve animals were administered NOX4i to evaluate the effect on mitochondrial function. State 3 respiration in presence of adenylates was determined for each mitochondrial complex activity, respectively. (**A**–**D**) CI-CIV activity was normalized to mitochondrial protein. * for *p* ≤ 0.05, ** for *p* ≤ 0.01, *** for *p* ≤ 0.001, and **** for *p* ≤ 0.0001.

**Figure 6 antioxidants-13-00489-f006:**
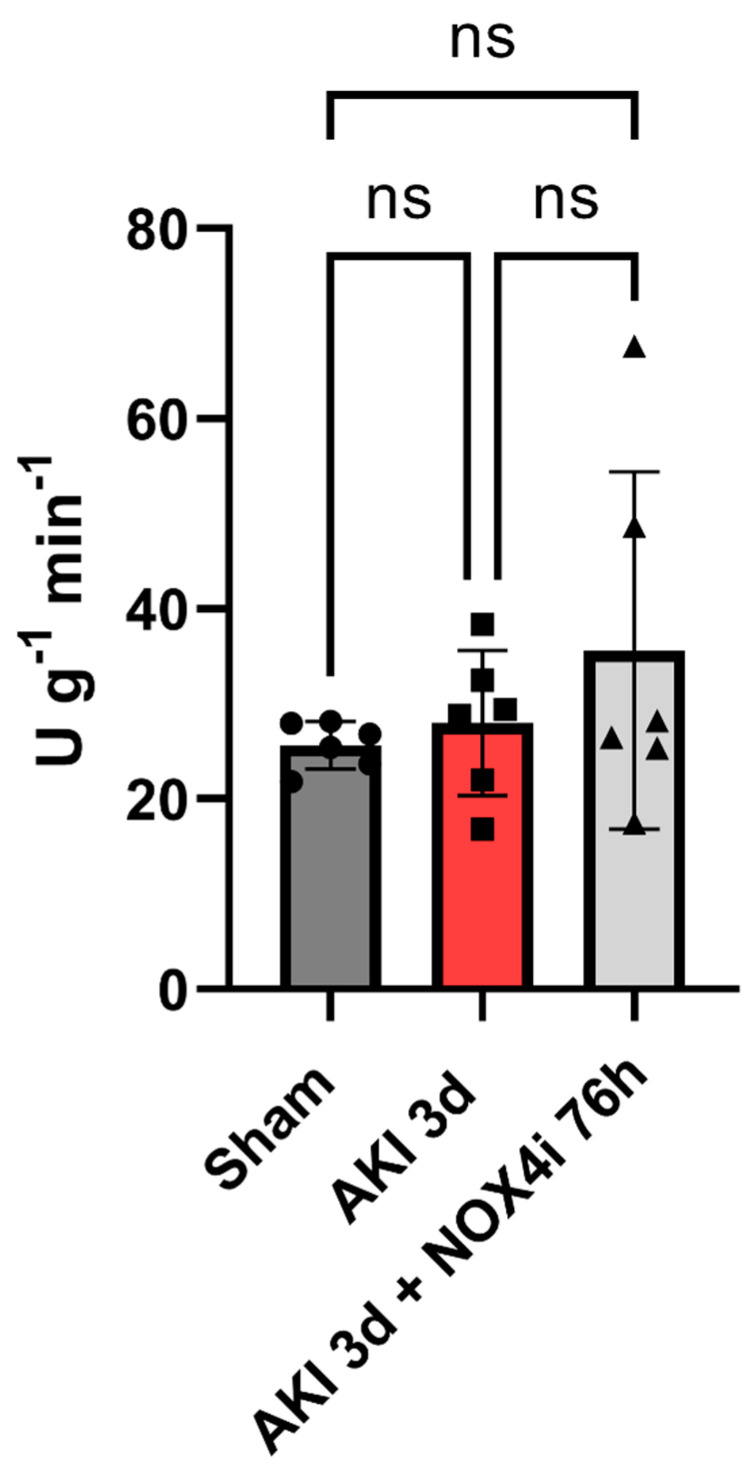
Kidney tissue citrate synthase activity was measured in mice after the induction of acute kidney injury (AKI) by 20 min ischemia followed by treatment with NOX4 inhibitor (NOX4i) during the reperfusion phase (N = 6/group except for AKI 3d + NOX4i 76 h, N = 5). Citrate synthase activity was measured by using a commercial colorimetric kit. One-Way ANOVA was used for statistical analysis. Non-significant variations were indicated as ns.

**Figure 7 antioxidants-13-00489-f007:**
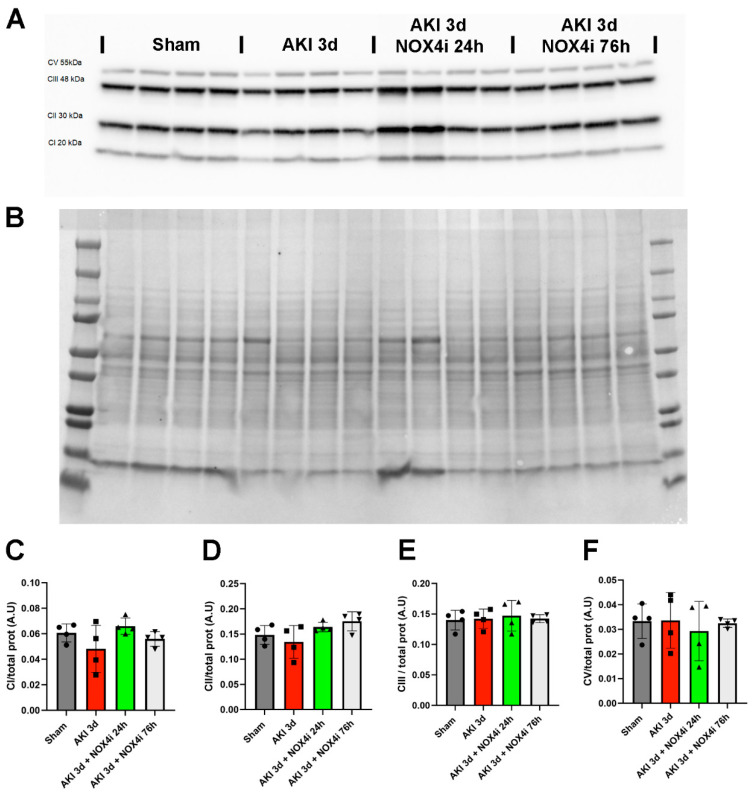
Immunoblotting was performed to evaluate the protein levels of the mitochondrial respiratory complexes in kidneys on the 3rd day of reperfusion (AKI) in animals treated with the NOX4 inhibitor (NOX4i) (N = 4/group). (**A**) Protein expression levels using a cocktail of antibodies targeting mitochondrial complexes. (**B**) Protein staining of the membrane used for normalization. (**C**–**F**) Statistical analysis of the protein levels of CI, CII, CIII, and CV (One-Way ANOVA).

**Figure 8 antioxidants-13-00489-f008:**
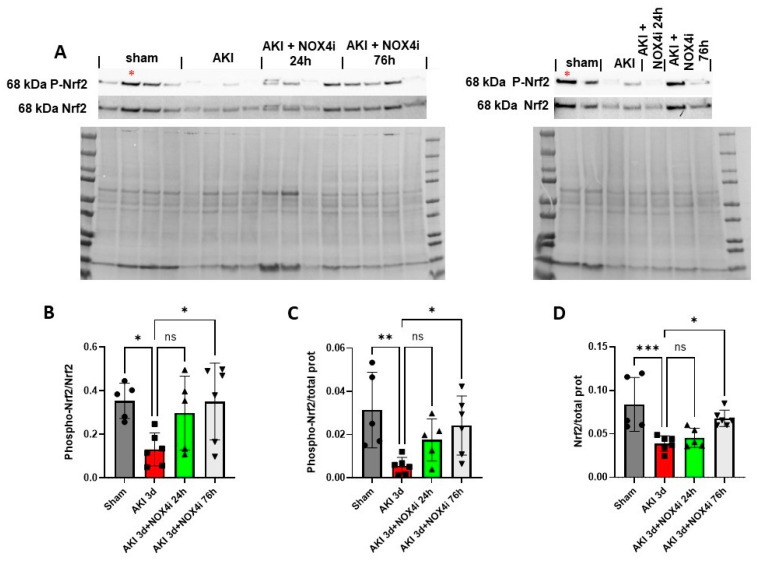
Immunoblotting was performed to evaluate protein levels of Ser40 phosphorylated Nrf2 in kidneys on the 3rd day of reperfusion (AKI) in animals treated with NOX4 inhibitor (NOX4i) (Sham N = 5, AKI 3d N = 6, AKI 3d + NOX4i 24 h N = 5, AKI3d + NOX4i N = 6). (**A**) Protein expression using antibodies targeting phosphorylated Nrf2 (Ser40), Nrf2, and membranes stained for protein used for normalization. Corrections between blots were made using a control sample denoted *. (**B**) Statistical analysis of the levels of phosphorylated Nrf2 related to Nrf2 and (**C**) total protein. (**D**) Nrf2 normalized to total protein. One-Way ANOVA was used for statistical analysis. * for *p* ≤ 0.05, ** for *p* ≤ 0.01, and *** for *p* ≤ 0.001. Non-significant variations are indicated as ns.

**Figure 9 antioxidants-13-00489-f009:**
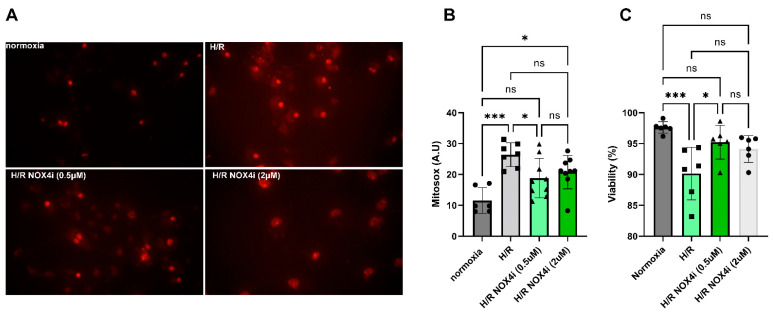
Mitochondrial ROS production was measured using MitoSOX™ in HK-2 cells exposed to hypoxia (0.2% O_2_) for 3 h and reoxygenated for 24 h (H/R) in the presence of either 0.5 or 2 µM NOX4i and subsequently analyzed under fluorescent microscopy. (**A**) Representative photos of MitoSOX-treated cells. (**B**) Statistical analysis of mitochondrial ROS production. (**C**) Trypan blue was used to evaluate the fraction of viable cells. * for *p* ≤ 0.05, and *** for *p* ≤ 0.001. Non-significant variations are indicated as ns.

**Figure 10 antioxidants-13-00489-f010:**
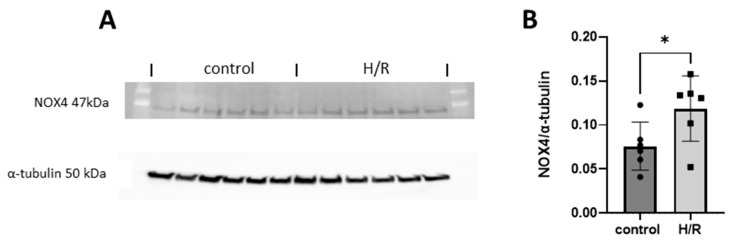
(**A**) Immunoblotting by targeting NOX4 was performed on HK-2 cells exposed to hypoxia (2% O_2_) for 3 h followed by reoxygenation for 24 h (H/R). (**B**) Statistical analysis of the NOX4 protein levels normalized to α-tubulin by using Student’s *t*-test. * for *p* ≤ 0.05.

**Figure 11 antioxidants-13-00489-f011:**
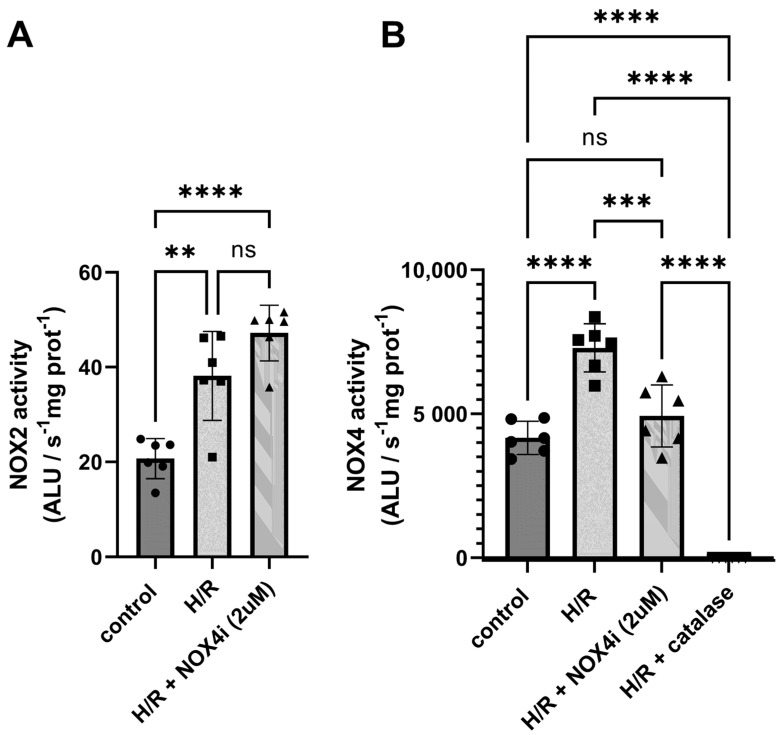
Cells were exposed to hypoxia (0.2% O_2_) for 3 h followed by reoxygenation for 12 h (H/R). (**A**) Superoxide production by NADPH oxidase was evaluated luminometrically in whole cells using lucigenin as a superoxide probe after supplementing with NADPH. (**B**) H_2_O_2_ production by NADPH oxidase was measured in sonicated cells by using luminol as a H_2_O_2_ probe after supplementing with NADPH in presence of HRP. One–Way ANOVA was used for statistical analysis. ** for *p* ≤ 0.01, *** for *p* ≤ 0.001, and **** for *p* ≤ 0.0001. Non-significant variations are indicated as ns.

## Data Availability

The raw data supporting the conclusions of this article will be made available by the authors on request.

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
