# Peer review of "Specific NOX4 Inhibition Preserves Mitochondrial Function and Dampens Kidney Dysfunction Following Ischemia–Reperfusion-Induced Kidney Injury"

_antioxidants, 2024, doi:10.3390/antiox13040489_

Round 1

Reviewer 1 Report

This is a well worked out study, the results section need some additional information. The discussion is quite comprehensive and well structured, and in a way to identifies additional mechanistic studies that will be quite useful in defining the mechanistic basis of NOX4 role in AKI of different etiologies. My comments and concerns are listed in the Detailed Comments Section..

1. Line 60- Please define DAMP.

2. Line 69- Please correct the font size.

3. Lines 87- 88- The sentence should read “---, to investigate the hypothesis that this treatment----.”

4- Line 251- Use the right size font and remove the underlining for “chemiluminesence.”

5- Please move the sentence starting on line 114 and ending on line 116 to the beginning of the paragraph.

6- Showing the localization of NOX4 in kidney post IR injury is needed to confirm its localization to the site of injury.

7- Figures 1, 2 and 3. What was the time point for AKI bar? The authors need to show the measurements for time matched AKI samples for both 24h and 76h samples. This is important because the AKI induced tubular damage is in a dynamic phase and the 24 and 76 hours post injury can present with different renal function and histopathologcal findings.

8- Please include scale bars for all figure panels and define the magnification of all figures. Also, the figures are very pixeled and need to be of much higher resolution. Please define the site of tubular damage and the type of injury (use arrows to show the injury types such as tubular casts, epithelial cell damage, tubular dilatation etc.). Why was a non-parametric statistical test (Kruskal-Wallis test) used for statistical analysis of the tubular injury scores in figure 2? Please make sure that the correct test is listed in the Methods Section and Fig. 2 legend.

9- Please add the data for mitochondrial assays for 24 hours in figures 4 and 5. This is needed since it is highly likely that the values at 24 hours post injury are not the same as those of 76 hours post injury.

10- I assume the AKI is at 76h in figure 6.

11- If possible showing the NRF2 immunolocalization would be helpful.

12- Figure 9C- Was the viability of cells at 2uM of inhibitor not significantly different from H/R cells?

13- Lines 398-399- Is this sentence the title for section 3.10.

14- Line 461- Please define FCCP and its uncoupling function.  

Author Response

  1. Line 60- Please define DAMP.

Thank you for bringing this to our attention. The issue has now been addressed in the manuscript.

  1. Line 69- Please correct the font size.

All font-related issues in the paper have been rectified.

  1. Lines 87- 88- The sentence should read “---, to investigate the hypothesis that this treatment----

This is now corrected.

4- Line 251- Use the right size font and remove the underlining for “chemiluminesence.”

  This has now been corrected

5- Please move the sentence starting on line 114 and ending on line 116 to the beginning of the paragraph.

We appreciate you highlighting this matter. It has been resolved in the manuscript.

6- Showing the localization of NOX4 in kidney post IR injury is needed to confirm its localization to the site of injury.

Thank you for your input on this matter; it would have provided valuable insights. However, we did not conduct this experiment as earlier studies demonstrated that NOX4 is present in the mitochondria, endoplasmic reticulum, and nucleus. (PMID: 20523116, PMID: 20185797, PMID: 20713697, PMID: 16324151).

7- Figures 1, 2 and 3. What was the time point for AKI bar? The authors need to show the measurements for time matched AKI samples for both 24h and 76h samples. This is important because the AKI induced tubular damage is in a dynamic phase and the 24 and 76 hours post injury can present with different renal function and histopathologcal findings.

We apologize for any confusion that may have arisen. To clarify, all AKI-groups were observed at the 76-hour (3-day) time point. The designation of the 24-hour group indicates that these subjects received the NOX4 inhibitor during the initial 24 hours of the 76-hour reperfusion period, as previously indicated in the legend of figure 2 and the method section. We have proceeded to include this clarification in all figures and legends to prevent any further misunderstandings.

8- Please include scale bars for all figure panels and define the magnification of all figures. Also, the figures are very pixeled and need to be of much higher resolution.

The recommended corrections have been addressed.

 Please define the site of tubular damage and the type of injury (use arrows to show the injury types such as tubular casts, epithelial cell damage, tubular dilatation etc.).

This information has now been included in both the figure and its legend.

Why was a non-parametric statistical test (Kruskal-Wallis test) used for statistical analysis of the tubular injury scores in figure 2?

We opted for a non-parametric test because the discrete values ranging from 1 to 5 did not conform to a normal distribution. While a standard One-way ANOVA indicated significant differences among “all” groups, it was not the appropriate test in this setting.

Please make sure that the correct test is listed in the Methods Section and Fig. 2 legend.

Thank you for observing this discrepancy. This is now corrected.

9- Please add the data for mitochondrial assays for 24 hours in figures 4 and 5. This is needed since it is highly likely that the values at 24 hours post injury are not the same as those of 76 hours post injury.

We kindly refer to the earlier comment indicating that all groups were sacrified 76 hours post-reperfusion.

10- I assume the AKI is at 76h in figure 6.

That is correct and now clarified in all figures

11- If possible showing the NRF2 immunolocalization would be helpful.

We acknowledge that including that information would have provided valuable insights. Regrettably, we did not perform this analysis as investigating Nrf2 activation was not a part of our original plan.  Nonetheless, the phosphorylation of Ser40 serves as a fairly reliable indicator of its translocation into the nucleus. We hope that you find this sufficient at the current stage.

12- Figure 9C- Was the viability of cells at 2uM of inhibitor not significantly different from H/R cells?

Indeed, they were not significantly different from either the control or H/R cells. As previously mentioned in the Results section, this is now also emphasized in the statistical graph.

13- Lines 398-399- Is this sentence the title for section 3.10.

Correct observation. The issue has now been rectified.

14- Line 461- Please define FCCP and its uncoupling function.  

This is now corrected

Reviewer 2 Report

This study presents data from a novel NOX4 inhibitor.  While this does provide useful and interesting knowledge to the scientific community, there is much that needs to be addressed.

Western blot figures should be cleaned up and combined, please look at other papers with western blots for comparison.  Please describe the normalization in the methods in greater depth. Fig 7 Why are the total protein and complex western blots not the same size? Molecular weight markers should be indicated on the complex blot. It looks like all CII and CIII bands were decreased in the AKI groups compared to sham.  Fig 8 Please repeat this blot with all samples together on the same membrane.  You CANNOT combine densitometry data from multiple blots without appropriate normalization to control bands.

 Please explain/give references to validate the cell culture IR model.  What were the number of technical and experimental replicates for the cell hypoxia experiments?  Experiments with cells should be performed on 3 separate days multiple times. All images need scale bars and the contrast should be improved. Representative images do not match the data in the graph.

 Why is tubulin used instead of total protein in Fig 10.  Please repeat and be consistent with your choice of WB normalization. Visually, there does not appear to be a significant difference in the NOX4 expression between groups. What is H/R?  Please be consistent with terminology/abbreviations and define all abbreviations in the figure legends as well as the text (eg. H/R vs hypoxia reox).

Fig 4A, Naiva = Naïve?

Fig 5 – Redundant, can use E-H and mention in the results text that complex activities were also assessed by normalization to tissue wet weight and had the same results.

Author Response

This study presents data from a novel NOX4 inhibitor. While this does provide useful and interesting knowledge to the scientificcommunity, there is much that needs to be addressed.

Western blot figures should be cleaned up and combined, please look at other papers with western blots for comparison. Please describe the normalization in the methods in greater depth.

We apologize for any misunderstanding regarding your comment on cleaning up the western blot figures. They might appear somewhat untidy compared to other papers because of the normalization to total protein, whereas others typically utilize a control protein for normalization. We kindly ask for your understanding as this discrepancy does not impact the primary message conveyed in the paper. The normalization method is now stated under section 2.10.

Fig 7 Why are the total protein and complex western blots not the samesize?

The blots, graphs and total protein membranes were not integrated in the same figure since we wanted to increase the flexibility for the journal to to rearrange them as needed. However, this led to the possibility that inadvertently adjusting the size by dragging one of the corners. To prevent this from happening, we have now consolidated all figures into one cohesive image, eliminating the possibility of such changes.

Molecular weight markers should be indicated on the complex blot.

We are slightly puzzled by this comment, as the molecular weight was indicated on the blot. If the suggestion is to merge the colorimetric blot with the luminometric (which is done for correct size analysis, and now attached here for you), it results in a compromised appearance of the bands/background due to the overlay. We have now included details about the protein ladder used, allowing identification of specific band sizes at the total protein blots.

 It looks like all CII and CIII bands were decreased in the AKI groups compared to sham.

Maybe for the eye, and there are a few dots that are lower but we trust the software analysis when normalized to total protein. To verify, we re-ran the analysis, and it yielded consistent results. 

 Fig 8 Please repeat this blot with all samples together on the same membrane.

You CANNOT combinedensitometry data from multiple blots without appropriate normalization to control bands.

In consideration of the lack of gels in our possession with an adequate number of wells, we divided the samples into two gels, with a control sample loaded in both. To account for variations between blots concerning P-Nrf2, Nrf2, and total protein, we adjusted by a calculating factor and applied this "difference" factor to the remaining samples. Your feedback was greatly valued, as it revealed an error in our calculations with regard to one of the factors and in addition, the control sample was inadvertently duplicated as two separate samples. Fortunately, this oversight did not impact the final results. The issue has been rectified, and a comprehensive explanation has been incorporated into the paper. We hope that this response addresses your concerns adequately.

Please explain/give references to validate the cell culture IR model.

The IR-model for the cells was developed in several steps as 1% O2 24h normally used for this cell-line had minor effects on viability and mitochondrial ROS production in our setting. To address this, we have included a note regarding this in the method section.

What were the number of technical and experimental replicates for the cell hypoxia experiments? Experiments with cells should be performed on 3 separate days multiple times.

Correct, we verified the results during 3 separate days of experiments. We have added this information to the manuscript

All images need scalebars and the contrast should be improved.

The necessary adjustments have been made to the manuscript.

Representative images do not match the data in the graph.

We have now carefully selected alternative images from the results that better align with the graphs, both for the TUNEL assay and the histopathological evaluation.

Why is tubulin used instead of total protein in Fig 10. Please repeat and be consistent with your choice of WB normalization.

Regrettably, during the course of these experiments, as we at the time did not have access to the protein staining agent, we had to resort to utilizing anti-alpha tubulin antibodies for normalization purposes. We kindly ask for your understanding as these results only serves as a verification of results observed by several other research groups.

Visually,there does not appear to be a significant difference in the NOX4 expression between groups.

Yes, we agree that visually it does not look to be a big difference. The significant difference appeared after the densiometry analysis using the software.

What is H/R? Please be consistent with terminology/abbreviations and define all abbreviations in the figure legends as well as the text (eg. H/R vs hypoxia reox).

We appreciate your input on this matter. As a result, we have ensured consistency in the terminology used throughout the paper.

Fig 4A, Naiva = Naïve?

Thank you for bringing attention to this misspelling; it has been corrected in the figures and text.

Fig 5 – Redundant, can use E-H and mention in the results text that complex activities were also assessed by normalization to tissuewet weight and had the same results.

According to your suggestions, we have now removed the results related to tissue wet weight and only mention it in the text.

Reviewer 3 Report

In this manuscript, the authors evaluated the efficacy and mechanism of NOX4 inhibitor (GLX7013114) in the treatment of ischemia-reperfusion (IR)-induced kidney injury. The results showed that chronic drug-specific inhibition of NOX4 could protect mitochondrial and renal function by reducing ROS and increasing Nrf-2 protein expression. In my opinion, this manuscript can be considered for publication in Pharmaceutics after minor revisions.

1. In the evaluation of renal function, creatinine concentration is as important as blood urea nitrogen, and authors should provide corresponding data.

2. The authors demonstrated that NOX4 inhibitors preserve mitochondrial function by reducing ROS in cells, but not in kidney tissue.

3. In the “3.7. Nrf2 serine residue 40 phosphorylation” part, the authors can provide some background to illustrate the role of Nrf-2 protein to enhance the reader's understanding.

4. Inflammatory factors also play a role in aggravating the condition during the occurrence of AKI. It is suggested that the authors also provide some data to illustrate.

5. There are some formatting errors in the manuscript, please check it carefully. For example, the size of part of the text is different, and the number unit in the picture is not uniform.

6. The most recent references cited by the authors are woefully inadequate. Related studies about AKI therapy are suggested to refer (J. Colloid Interf. Sci., 2024, 662, 413-425; Acta Biomater., 2024, 173482-494Theranostics, 2023, 13, 2863-2878.).

Author Response

In this manuscript, the authors evaluated the efficacy and mechanism of NOX4 inhibitor (GLX7013114) in the treatment of ischemia-reperfusion (IR)-induced kidney injury. The results showed that chronic drug-specific inhibition of NOX4 could protect mitochondrial andrenal function by reducing ROS and increasing Nrf-2 protein expression. In my opinion, this manuscript can be considered forpublication in Pharmaceutics after minor revisions.

Detail comments

  1. In the evaluation of renal function, creatinine concentration is as important as blood urea nitrogen, and authors should provide corresponding data.

We appreciate your feedback. Given that mice, unlike humans, actively secrete creatinine (PMID: 20032962), we opted to utilize BUNs and actual GFR measurements using FITC inulin injections as a significantly more precise method for estimating GFR. We hope that you think these measurements are sufficient at the current stage.

authors demonstrated that NOX4 inhibitors preserve mitochondrial function by reducing ROS in cells, but not in kidney tissue.

This is a highly valid comment. We attempted to assess NADPH-oxidase activity in kidney tissue homogenates following our observations in the cell experiments. Unfortunately, we encountered challenges in measuring NOX activity in tissue homogenates, as we either obtained no signal or very weak signals. Had the experiments been successful, we would have certainly incorporated them into the paper.

  1. In the “3.7. Nrf2 serine residue 40 phosphorylation” part, the authors can provide some background to illustrate the role of Nrf-2protein to enhance the reader's understanding.

We have now added a brief explanation in this section with regard to the relationship between Nrf2, NOX4 and mitochondrial health.

  1. Inflammatory factors also play a role in aggravating the condition during the occurrence of AKI. It is suggested that the authors also provide some data to illustrate.

Your input is valid, and we evaluated various inflammation markers including TNF-alpha, KC, and IL-6. Regrettably, the results did not provide conclusive findings, leading us to decide against including this information in the paper.

  1. There are some formatting errors in the manuscript, please check it carefully. For example, the size of part of the text is different, and the number unit in the picture is not uniform.

Thank you for noticing this. It is now corrected throughout the paper

  1. The most recent references cited by the authors are woefully inadequate. Related studies about AKI therapy are suggested to refer

(J. Colloid Interf. Sci., 2024, 662, 413-425)

This paper investigate treatment involving scavenging of ROS and nitrogen species delivered through nanoparticles. We choose not to use this reference since our focus was not on ROS scavenging. Including such reference would necessitate citing numerous additional sources in this particular field. We hope you appreciate and understand the rationale behind this decision.

Acta Biomater., 2024, 173, 482-494  See above comment

Theranostics, 2023, 13, 2863-2878. This reference was already included in the original manuscript (lines 75-76).

Submission Date

04 March 2024

Date of this review

21 Mar 2024 08:09:20

© 1996-

We were uncertain about the specific review article mentioned in your comment.. If it pertains to the manuscript with the PMID: 38541160, it has now been included in the paper.

Round 2

Reviewer 1 Report

I thank the authors for their through response and agree that the manuscript is much improved. 

My point on showing the localization of NOX4 in kidney post IR injury in the kidney to confirm its localization to the site of injury still stands. While the references provided by the authors show the differential expression of NOX4 in a variety of tissues including cardiomyocytes the cell types and injuries are qualitatively different than AKI. I understand that the authors may not have the means to address this question; however, the absence of localization of NOX4 in injured kidney and where authors think NOX4 expression in may be enhanced in the injured kidneys (e.g., epithelial cells of the S3 segment of PT) should be discussed in the discussion section. A similar discussion on the localization of NRF2 and p-NRF2, a transcription factor with important protective function, would be appropriate.  

Author Response

My point on showing the localization of NOX4 in kidney post IR injury in the kidney to confirm its localization to the site of injury still stands.  While the references provided by the authors show the differential expression of NOX4 in a variety of tissues including cardiomyocytes the cell types and injuries are qualitatively different than AKI. I understand that the authors may not have the means to address this question; however, the absence of localization of NOX4 in injured kidney and where authors think NOX4 expression in may be enhanced in the injured kidneys (e.g., epithelial cells of the S3 segment of PT) should be discussed in the discussion section. A similar discussion on the localization of NRF2 and p-NRF2, a transcription factor with important protective function, would be appropriate.

We acknowledge and value the significance of highlighting the protein localization to enrich the paper's quality. As previously noted, our capacity to retrospectively perform these experiments is limited. We aimed to steer clear of speculative statements regarding the localization and have now instead addressed this as a study limitation in the discussion section. We hope that this clarification meets your expectations.

Reviewer 2 Report

 Figure 8.  I appreciate that you took into consideration the normalization for combining multiple blots; however, there is quite a lot of variability between all 6 sham animal lanes, which can allow for data to be cherry picked.  Instead of using one band, you can take the average intensity of all control lanes after normalization to loading and then normalize all bands to that value.  Thus you have effective relative difference relative to the control mean.

NOX4i Solvent and control osmotic minipumps need to be included in the methods

 The number of mice analyzed from each group need to be reported in the figure legends

 No Na or Cl transport or urine concentrations were measured, so you cannot say that the inhibitor likely improved this in the discussion unless you measure and report those values (paragraph 3)

None

Author Response

Figure 8. I appreciate that you took into consideration the normalization for combining multiple blots; however, there is quite a lot of variability between all 6 sham animal lanes, which can allow for data to be cherry picked. Instead of using one band, you can take the average intensity of all control lanes after normalization to loading and then normalize all bands to that value. Thus you have effective relative difference relative to the control mean.

We appreciate your suggestion, but unfortunately, we believe there may have been a misunderstanding regarding our calculations for normalization between membranes. The presence of an asterisk (*) marking the same animal/sample in both distinct blots signifies a deliberate choice without bias, showcasing the varying transfer efficiency across the membranes.  Assuming uniform transfer efficiency for the rest of the samples in membrane 2 (similar to the control sample marked with *), the correction factor that appeared was used for the remaining samples. Attempting to average the sham samples and correct them based on the averages of different controls in membrane 2, (with the exception of the identical control sample) could lead to a less precise estimation of transfer efficiency. We have now given an even more extensive explanation in the paper how the normalization was performed.

NOX4i Solvent and control osmotic minipumps need to be included in the methods

The provided information has now been included in the method section.

The number of mice analyzed from each group need to be reported in the figure legends

This is now included in the figure legends

No Na or Cl transport or urine concentrations were measured, so you cannot say that the inhibitor likely improved this in the discussion unless you measure and report those values (paragraph 3)

We appreciate your observation. The speculation has been duly removed from this paragraph.